# An American Notwithstanding Clause? Between *Potestas* and *Potentia*

**Boleslaw Z. Kabala** [1,2,*] **and Rainey Johnson** [3]

1   School of Humanities and Social Sciences, College of Undergraduate Studies, Colorado Christian University, Lakewood, CO 80226, USA
2   Department of Social Sciences, College of Undergraduate Studies, Colorado Christian University, Lakewood, CO 80226, USA
3   Department of Government, Legal Studies, & Philosophy, Tarleton State University, Stephenville, TX 76401, USA; rainey.johnson@go.tarleton.edu
*   Correspondence: bkabala@gmail.com

**Abstract:** Debates about judicial review and departmentalism have continued to rage, and in the wake of the last three Supreme Court appointments and current Presidential Commission on the Court, only look to intensify. Should the US adopt a notwithstanding or override provision, of the kind that exits in Canada and Israel? These countries take a departmentalist approach to allow the legislature to override the Court, "notwithstanding" its ruling. Although America is a presidential framework, a paradox emerges: evidence exists that its system already makes possible the equivalent of a notwithstanding clause. This consists of Congress and the President together "overruling" the Supreme Court. In another sense, however, this is not an accepted practice—large parts of the legal community hold that the US Constitution establishes judicial supremacy. To better understand this dynamic, we consider two kinds of power: formal and authorized (potestas) as well as direct and concrete (potentia). The contrast between the positions on both power and sovereignty of Thomas Hobbes (associated with potestas) and Baruch Spinoza (linked to potentia) helps clarify these issues in a contemporary context. It turns out that a robust departmentalist equivalent of the notwithstanding clause already exists in the US, as a matter of Hobbesian potestas but not of Spinozist potentia. Another term for the latter is pouvoir constituant. Spinoza's perspective on political activity further clarifies the in-between nature of the American override capacity: the active or passive character of a multitude is not binary, but is a matter of degree. Without making an institutional recommendation, we note that Spinoza's understanding of power also allows for dynamic interaction between potentia and potestas: formal authorization can contribute to the expression of direct power. It is, therefore, conceivable that additional codification of the existing American override capacity, either through a joint declaration of Congress and the Presidency or a Constitutional Amendment, can strengthen the effective sovereignty of the American people in relation to the courts.

**Keywords:** *potestas*; *potentia*; notwithstanding clause; Hobbes; Spinoza; departmentalism; judicial supremacy; pouvoir constituant; jurisdiction stripping

## 1. Introduction

With the Presidential Commission on the Supreme Court meeting, and after the three successful appointments of the Trump presidency, questions related to court reform have reached a fever pitch. Of course, these debates are not new. The Reconstruction era saw legislative supremacy as a Congress that included numbers of radical Republicans changed the constitution of our nation's highest tribunal and deprived it of the ability to hear important cases. The tumultuous events of the 1930s, leading to what is sometimes casually referred to as the "switch in time that saved nine", obscure the fact that the back and forth between President Roosevelt and Congress reached proportions of a full-blown

constitutional crisis. Today, however, given the dissatisfaction of both right and left with judicial overreach, why not consider a formal override or notwithstanding clause?

The countries known to have this kind of system include Canada, Israel, Germany, and New Zealand. Inevitably, the question arises: if this degree of democratic accountability can work for them, why would it not be the case, potentially, that it could serve popular responsiveness in the United States? Could a "legislative veto" of the Supreme Court not diffuse our own disagreements about whether to expand the size of the Supreme Court?

Paradoxically, we find significant evidence in the US that a measure of override power both already exists, and that it does not. The US is not a parliamentary government, so comparisons to Israel and Canada are tricky. However, Congress can pass as a bill, and the President can sign into law measures that definitely do *not* allow the pronouncements of the Supreme Court to have final sway. One example is jurisdiction-stripping, which continues to occur and consists of the House and Senate, joined by the President, denying the Supreme Court the ability to hear a case or category of cases. At the same time, the consensus, or near consensus, of legal experts is that judicial supremacy is the law of the land. Is it possible to clarify the situation?

As it turns out, these debates are connected to deeper issues of power and sovereignty, with at least two competing conceptualizations. One is potestas, or authorized power. The other is potentia, or directly applied and concretized power. Additionally, this contrast is especially well illustrated by the disagreement between Hobbes and Spinoza on precisely these questions of what sovereignty involves. Hobbes, in constructing government, supports authorized or formal power. Spinoza, in disagreement with him, proposes sovereign influence that is concrete and direct. As will become clear, both of these forms of power are important to keep in mind to better understand the US legal landscape.

First, we provide background on the Canadian and Israeli systems of legislative override. We show that the Canadian veto is older and, in fact, more far-reaching. The Israeli framework applies only to economic liberties, whereas the former has been embroiled in more deep-seated cultural controversies. Although additional examples of weak judicial review or legislative supremacy could be provided, the Canadian and Israeli cases are both interesting, having transitioned from a system of intended or actual strong judicial review, to one where the people are able to overrule the courts.

Second, we note that, although the United States does not have a notwithstanding clause spelled out in its Constitution, important elements of this kind of override power do effectively exist. In addition to jurisdiction stripping grounded in Article III, there has also been court abolition in US political history; additionally, alongside these examples, there is the ongoing struggle between Congress and the Court related to the Religious Freedom Restoration Act, in which the national legislature has not hesitated to assert its prerogative of constitutional interpretation. However, to the extent that Cooper v. Aaron is a near consensus, especially among American legal academics, it is supremacy that reigns. Given the contradictory evidence, a step back is necessary. How is it that the power of override in the United States appears to both exist and to not exist at the same time?

Third, and for the sake of better understanding this dynamic, we unpack two different conceptualizations of power. These are potestas and potentia. Potestas is authorized sovereign control or influence on paper, whereas potentia is its direct, concrete, and immanent application. This framework helps: if potestas is understood to broadly authorize jurisdiction stripping, it is possible to see how it supports departmentalism. However, if potentia refers to the direct exercise of sovereign control, it is possible to appreciate how this authorization does not correspond to exercise on the ground. Practice in the US has seemed to reflect the ideal of judicial finality, and therefore of the power of judges as potentia, outside cases in which the opposite seems true and the concretized power of the people does seem to prevail (in jurisdiction stripping and court abolition), but in which, as defenders of judicial supremacy have pointed out, the stakes are not high.

As it turns out, the two thinkers most relevant to enhancing understanding of this difference between potestas and potentia are Thomas Hobbes and Baruch Spinoza. Hobbes

is typically associated with potestas; it is Spinoza who is viewed as theorizing potentia. Intriguingly, both, in their later works, moved closer to the relational and dynamic dimension of potentia or applied power. This view also points to the important concept of pouvoir constituant. Additionally, just as he is with respect to potentia, Spinoza generally is viewed as more supportive of pouvoir constituant, or the direct impact of the people in their originary state on a political outcome. Interestingly, despite the direct relevance of this concept from the revolutionary tradition to the possibility of abolishing courts, a vanishingly small percentage of articles dealing with both judicial supremacy and departmentalism mention pouvoir constituant.

Fourth, and especially helpful in making this case, are the different treatments of judicial institutions in both these authors. The way Spinoza theorizes the courts in his *Political Treatise* especially speaks to and is supportive of departmentalism. Unlike Hobbes, Spinoza is not open to a system of judicial nomination. Spinoza says explicitly that judges are chosen from the ranks of parliamentarians; furthermore, their number must be high and they are strictly term limited. Hobbes is as far from judicial finality as is Spinoza. However, the way in which he sets up his judicial offices makes it more likely, someday, than an assertion of supremacy from this element of the polity will come. This is analogous to how *actual* bodies in the Hobbesian commonwealth, as Sandra Leonie Field has shown, co-opt civil sovereignty.

Fifth, we show that there is an additional mode in which Spinozist metaphysical categories add clarity to these debates. In the US, given that the override provision seems to exist effectively in some areas but not in others, one can understand confusion related to whether the power exists in the first place. One response is two kinds of sovereign influence. However, Spinoza's helpful "metaphysics of degree" provides an additional way around this dilemma. It is not that the power of override does, or does not, exist. *Rather, it is that it exists to a degree.* Thus, for Spinoza to be active in a technical sense means to be in possession of right. Additionally, "possession of one's rights" applies not only to individuals, but to bodies and institutions as well. Now, from the "power as potentia" perspective of pouvoir constituant, the active multitude that has maintained its right is able to overthrow and reconstitute courts at will. However, this important category of not having relinquished the right does not exist for Spinoza as a simple binary. The question of an active multitude, for Spinoza, is one of degree.

Sixth, and finally, we consider that potentia and potestas as conceptualizations of power are not mutually exclusive. A constitution, of course, is a form of authorized power. Hobbes does not rely on one in *Leviathan*, even as he emphasizes the importance of juridical considerations that include the contract. How about Spinoza? As it turns out, he is open to constitutionalism, and especially so in his book, the *Political Treatise*, where potentia is emphasized most. Formal authority or potentia may not correspond to influence on the ground. However, if "influence on the ground" is real enough to be able to express itself in a written document, this reaffirms its existence. Additionally, it is possible that power maintains its mode longer with written reification. Potestas, in other words, can increase potentia. Without, therefore, taking sides—as, indeed, we also present reasons *not* to move further in a departmentalist direction—we emphasize that people who favor more jurisdiction stripping and court abolition may want to consider a joint declaration of Congress and the Presidency, or a constitutional amendment itself.

## 2. Parliamentary Systems/Canada and Israel

In a number of countries around the world, no judicial review mechanism is understood to provide a court or courts with final authority to strike down laws deemed inconsistent with its constitution. Canada, England, New Zealand, and Israel come to mind. Specifically, these are nations with a tradition of parliamentary supremacy. While there is some evidence that, although civil law systems do not historically preclude judicial review, a common law framework recognizing stare decisis can contribute independently to it (Colon-Rios 2014a, p. 147; Stephanopoulos 2005; de Andrade 2001).

Out of these countries with significantly weaker systems of review, two stand out: Canada and Israel. This is because both moved to adopt, or adopted at a specific time, *actual* judicial review that allows judicial bodies to declare laws *invalid*, as opposed to merely inconsistent with the constitution. They both subsequently added the equivalent of a notwithstanding clause, or formal mechanism whereby the legislature can veto the decision of the Supreme Court, seeming to take a step back towards legislative sovereignty.[1] They did so in radically different ways, however, and so it is important to provide a quick review.

The adoption of the Canadian clause goes back to the early 1980s and, in fact, to the moment of popular constitutionalism in 1982, in which Canada as a fully sovereign nation was born. That year saw the adoption of the Charter, which protected basic rights irrespective of parliamentary will and majority. It is important to remember that, before this year, there was no written constitution in Canada, and England was able to veto constitutional changes that Canada may have wished to make (Dodek 2016, p. 50). The Canadian Prime Minister at the time, Pierre Trudeau, had been pushing for patriation and a written constitution for some time.[2] However, many of the provincial governments were reluctant to surrender some of their power to the Charter, hence the notwithstanding idea that Alberta, Newfoundland, and Saskatchewan especially suggested in return for their support of and consent to said Charter.[3] This compromise, including a notwithstanding clause that allowed an override of the Charter by either the federal or provincial governments, came to fruition.

Did the notwithstanding clause apply to the whole Charter? No—and this is reflected in section 33 of the Charter itself. Both the federal Parliament and the provincial legislatures can "expressly declare by a simple majority that a law shall operate notwithstanding a provision" in section 2 (fundamental freedoms: conscience, religion, expression, assembly, etc.), sections 7–14 (legal rights: life, liberty, security, etc.), and section 15 (the equality rights: equal protection and equal benefit of the law). Strikingly, the provision did not cover all the crucial rights and freedoms guaranteed by the Charter. Mobility rights, education rights, democratic rights, and language rights were all left out of the decision. Overrides would expire after five years unless reenacted at that time.[4]

Controversy ensued almost from the beginning. The Quebec parliament voted to repeal every law passed before adoption of the Charter, and to pass it again with the notwithstanding provision attached. This omnibus use of the clause was struck down by the Quebec Court of Appeals, although it was overruled by the Supreme Court in Ford v. Quebec. This decision upheld the blanket application of the Charter's section 33 language, even as it struck down the idea that it could apply retroactively. It was then decided that, for the notwithstanding clause to be used, it must be done preemptively, with a formal mention of the Charter (Weinrib 1990). Again, in 1988, after a Quebec Bill prohibiting the use of English on commercial signs was struck down by the Supreme Court, Quebec chose to exercise their override right by reimplementing the ban on signs by enacting a notwithstanding clause. The resulting bill, though representing a minor compromise, allowed for the ban of English on commercial signs outside.[5] Today, the Canadian override language in the Constitution continues to polarize; Doug Ford, the Ontario Premier, recently relied on it to restrict election advertising by third parties. Additionally, Francois Legault, the Quebec Premier, has used it to defend secularism understood as limiting public displays of Muslim hijabs, Jewish kippahs, and Sikh turbans.[6]

The adoption of the Israeli clause, on the other hand, occurred later and was not the result of extensive public dialogue leading to the adoption of the first written national

---

1    (Colon-Rios 2014a, p. 147; Weill 2016, p. 121).
2    (Dodek 2016, p. 51).
3    (Stephanopoulos 2005).
4    Ibid. and see https://www.solon.org/Constitutions/Canada/English/ca_1982.html (accessed on 7 September 2021).
5    (Dodek 2016). Mulroney was angry with the decision and condemned the entirety of section 33, claiming: "any constitution that does not protect the inalienable and imprescriptible rights of the individual Canadians is not worth the paper it is written on" (p. 61).
6    See https://theconversation.com/doug-ford-uses-the-notwithstanding-clause-for-political-benefit-162594 (accessed on 7 September 2021).

constitution. The origins of the process are traceable to the period shortly after the end of the Second World War, when the Knesset, in its first meeting (1950), delegated to its Constitution, Law, and Justice committee the task of composing a Constitution piece by piece (Weinrib 2016, p. 69). This led to a series of what are known today as 13 Basic Laws, addressing subjects from the House of Representatives and Presidency themselves to the state of the economy and human rights, as well as budgetary and tax issues and the status of Jerusalem. To this day, there remains in Israel no single written constitution. The last two Basic Laws that were adopted, on Human Dignity and Liberty (2012) and Freedom of Occupation (1994), were modeled on Canada's Charter of Human Rights.[7]

In striking contrast to constitutional developments in Canada, in Israel, it was the Supreme Court that single-handedly, in the *United Mizrahi Bank* decision, claimed and articulated the power of judicial review for itself. It did so specifically by declaring the two most recent Basic Laws as constituting the actual written Constitution. This has widely been referred to as a constitutional revolution (Weill 2016) and, more specifically, as the equivalent of Marbury v. Madison in Israel.[8] However, in a significant way, *United Mizrahi Bank* went beyond *Marbury:* the incoming President of the Court, Aharon Barak, "used the language of revolution rather than evolution to capture the essence of this legal development" (Weill 2020). In the words of an author soberly outlining the magnitude of the constitutional shock, "This is a deviation from previous judicially-led revolutions and marks his awareness of the historical moment" (Weinrib 2016, p. 69). Barak had already previously contributed to significant paradigm shifts in constitutional law related to the Court and public petitioners as well as issues related to justiciability, but extending judicial review to primary legislation was a more momentous step. Although the Canadian and Israeli cases have both produced a notwithstanding clause, the path leading to this institutional outcome was different in the two countries and, arguably, more revolutionary in Israel.[9]

Following this articulation of judicial review, the controversy in Israel that led to the adoption of the notwithstanding clause was the result of the Supreme Court striking down a statute passed by the Knesset banning the importation of non-kosher meat. The decision was *Meatrael Ltd. v Prime Minister and Minister of Religious* Affairs (1993), in which the judges ruled that the statute was inconsistent with the Basic Law on Freedom and Occupation.[10] Intriguingly, the notwithstanding clause was made to apply only to one of the 14 Basic Laws—on Freedom of Occupation. Since this controversy, in which the Knesset succeeded in passing the statute despite the ruling of the Court, the notwithstanding clause has not been invoked.

Despite this relative calm, as the Court in Israel has continued to strike down legislation it deems inconsistent with On Human Dignity and Liberty,[11] a debate has emerged over whether or not the Israeli Parliament should extend the clause's power beyond economic and occupational rights. Those in Israel in favor of extending the override to legislation that impacts human rights tend to be members of religious and nationalist parties, who are inclined to see the Court (especially in the wake of *United Mizrahi Bank* and subsequent judicial invalidations of statutes) as a modernizing and secularizing threat. The contrast with current US politics, where the equivalent of a notwithstanding clause (or outright abolition of judicial review) is supported by those on both the far left and right, is worthy of note.

### 3. Jurisdiction Stripping

Evidence for a US legislative override exists: jurisdiction stripping is real. In a number of instances, Congress has pushed back against the courts by simply denying

---

7    Ibid., p. 67.
8    Ibid., p. 110.
9    Ibid.
10   Ibid.
11   (Weill 2016, pp. 105–6).

them the ability to hear cases. It has done so based, first and foremost, on Article III of the Constitution, and specifically Section 1, according to which judicial power exists only in those courts, outside the Supreme Court, that the US Congress "may from time to time ordain and establish"; and Section 2, according to which Supreme Court appellate jurisdiction can be modified (Fallon 2010, pp. 1065–69). Significantly, Article III, in this way, indicates that all tribunals, besides the nation's highest, depend for their very existence on Congressional legislative enactment, and may, in fact, be discontinued by the same body. These questions gained in urgency during the Bush administration, which sought to limit the ability of courts to hear habeas corpus cases related to Guantanamo, but was denied the right to do so in Boudemiene.

One important body of literature refers to the view that Congress has broad authority to strip as the "orthodox view".[12] A significant part of this dialogue is profitably traced to Henry Hart's seminal 1953 article on the subject (Hart 1953). Remarkably, even as this towering legal thinker and co-founder, with Albert M. Sacks, of what became the legal process paradigm of law, found a broad right of jurisdiction stripping so long as people had access to the legal system *on some level*, others immediately pushed back to say that he had not deferred sufficiently to Congress. This included one of his collaborators, Herbert Wechsler (Wechsler 1965), as well as Gunther (1984) and Black (1975), Monaghan (2019), Van Alstyne (1973), Bator (1982), Redish (1982), Bradley and Siegel (2017) (A number of these articles are mentioned in Monaghan 2019, pp. 13–15), and Casto (1985, 1990). Their arguments are certainly contested (Clinton 1984). However, that it is the orthodox view is further supported by the fact that even those who dissent from in part, whose number includes Akhil Amar, accept its status (Amar 1985). An important legal textbook also refers to the assessment of Congress's general power to modify jurisdiction in this way; (Mentioned by Monaghan 2019, p. 15).[13] Additionally, the ability to represent an "orthodox view" is further confirmed by the fact that thinkers from across the ideological spectrum, from Robert George to Mark Tushnet, believe more pushback against the Courts is a constitutionally healthy option.

In the wake of Boudemiene, however, Richard Fallon has done especially fine-grained work pointing to the complexity of the questions involved (Fallon 2010). Depending on the context, it can be hard to know the extent to which the power in question exists. In addition to constitutional issues, there is simply a lack of significant test cases. Those cases that have occurred, from a constitutional perspective, are still relatively minor and on the periphery, see Sprigman (2020).[14]

This ambivalence goes beyond general considerations: Fallon is especially effective in detailing that potential restrictions apply differently to various kinds of courts. Thus, on a given issue, is Congress able to take away the power to hear an important category of cases from the Supreme Court (Fallon 2010)?[15] This would limit the Supreme Court's appellate jurisdiction, making the decisions of lower national courts binding. How about from either district and appeals courts, or both, but not from the Supreme Court?[16] It was Congress, after all, that gave national courts plenary jurisdiction over federal questions in 1875 (Fitzpatrick 2012, p. 840), so the idea does not seem so far-fetched. In this scenario, the Supreme Court, and/or appeals courts, would hear more cases on original jurisdiction. How about taking away judicial oversight power from the Supreme Court and all federal courts, but not from state tribunals?[17] Or removing it from the totality of courts at the state and federal levels?[18]

---

12  Ibid., pp. 1067–69, 1078.
13  (Strauss et al. 2017).
14  Ibid., p. 1045.
15  (Fallon 2010, pp. 1087–93).
16  Ibid., pp. 1093–95.
17  (Fallon 2010, pp. 1083–87).
18  Ibid., pp. 1095–98.

With parallels to Hart's work, who also found a significant space in which Congress could act, though with limitations, Fallon concedes that a core jurisdiction-stripping power of national legislature exists. However, it must be possible to vindicate bona fide constitutional rights *somewhere* in the court system. Fallon also argues, contra interpretations of McCardle, that intent behind the limitation matters; if Congress is restricting the ability to hear cases based on the motivation to deny rights, this is problematic.

However, that leaves many questions unanswered. First, what if there is a longstanding disagreement between the people and judges with respect to what is, and is not, a constitutional right? Second, even disregarding this disagreement, is the level of courts from which to strip jurisdiction a matter of preference, or is the question of ultimate tribunal able to hear a case a substantive one? A lack of ready answers to these questions makes it harder to say, straightforwardly, whether or not a jurisdiction stripping power exists.

Frustratingly, a similar diversity of contexts exists when it comes to remedies. These can include a writ of habeas corpus, compensation (economic damages), or injunctions (someone is ordered to desist). Can Congress *restrict* the ability of the court to make up for what somebody has lost, or say that a court simply cannot award damages, or provide injunctive relief?[19] Here, Fallon clarifies that, analogously to questions of *jurisdiction*, Congressional ability to take away a court's power to hand down remedies exists but is limited. It applies to those cases where a substantive right does not exist.[20] Additionally, if the right does exist, a remedy *somewhere* must be available in the court system, making possible some but not all stripping measures. As with jurisdiction questions, then, remedy issues point to the indeterminacy of this Congressional power, making it hard to know whether or the extent to which the power to jurisdiction strip exists.

Non-Article III courts further muddy the waters. These are legislative or administrative, with the first applying in a US territory and the second related to agencies and charged with adjudication on the question of whether a rule or regulation applies. Here, again, for Fallon, a core power to strip exists. However, it is even hazier than the analogous abilities of Congress to limit or take away jurisdiction, and the courts' awarding of damages, respectively, in Article III contexts. Fallon concludes that, generally, jurisdiction to conduct a de novo review of a decision made by a non-Article III judicial body (legislative or administrative court) can be taken away from any one particular Article III judicial body. However, *some* standing must be had in *another* Article III court if liberal democracy and the administrative state are to co-exist and survive.[21]

Even as they further complicate these questions, non-Article III Courts deepen the conversation in a fruitful way. The reason is that, regardless of whether the influential commentators who posit a broad jurisdiction stripping power are right or not, the administrative courts demonstrate that this practice does occur. The relevant paragraph, in which Fallon explains, is as follows:

> Today, non-Article III judges in administrative agencies and legislative courts vastly outnumber Article III judges and collectively adjudicate far more cases. Accordingly, the topic of Congress's power to substitute non-Article III federal adjudicators for Article III courts and to preclude judicial review of the non-Article III tribunals' decisions holds great importance.[22]

Involved here are not the big picture questions of flag burning or abortion that often come to mind when Congressional impact on jurisdiction is raised. However, these courts show definitively that the practice of stripping occurs.

Indeed, Dawn Chutkow has done work quantifying the extent to which it does (Chutkow 2008, p. 1058). Her results are as follows: since 1943, Congress has passed 248 laws and those contained 378 provisions that, in one way or another, deprived federal

---

19    Ibid., pp. 1101–15.
20    Ibid., p. 1104.
21    Ibid., pp. 1117, 1123.
22    Ibid., pp. 1115–16.

courts of review power. More interestingly, still, the *reason* in this context for jurisdiction restriction is not, as one might imagine, ideological. Rather, it is to head off legal action, which delays the implementation of policy and involves significant costs.

## 4. More Significant Jurisdiction Stripping

More significant and very recent cases of jurisdiction stripping exist as well, which do make it seem that an effective American notwithstanding clause is in place. Shockingly, one involved Congress passing a law to break off litigation *as it was being heard.* In Patchak v. Zinke, reminiscent in some ways of the Reconstruction-era McCardle and decided in 2018, David Patchak sued the Secretary of Interior Ryan Zinke over a residence next to a specific stretch of land. This was the Bradley property in Michigan. The background involved recognition by the Federal government, in 1999, of Match-E-Be-Nash-She-Wish, Native Americans of the Gun Lake Tribe, who immediately asked for and were granted their request to have the territory fall under the administration of the Indian Reorganization Act. This made possible a casino on the territory. Patchak argued that he would sustain multiple losses as a result. He also made his case on administrative grounds, given that the tribe was not officially recognized by Washington DC in 1934.

The district court held that Patchak had standing to sue the Secretary of Interior. After he did, Congress passed a bill that simply disallowed *any* suits related to the Bradley property. This was the Gun Lake Act of 2014. It was signed into law by President Obama.

A number of peer-reviewed articles characterize what happened as a serious violation of the separation of powers (Fisher 2017; Zoldan 2018). In describing the dynamics related to this case, Aziz Huq has also focused on the disconnect between ideal and reality of judicial independence, showcasing, in particular, Roberts' dissent from the Patchak plurality. For Roberts, who expressed grave concerns about the separation of powers in this context,[23] the decision by the plurality also legitimated singling out an individual by the legislature in a way that approximates a bill of attainder (Monaghan 2019).[24] Again, from this perspective, it would seem that the equivalent of an American notwithstanding clause is already in effect (Huq 2021), see Zoldan (2018) above.

## 5. Court Abolition

Entire courts have not only also had their jurisdiction modified but been abolished. This certainly happened in the early republic. In 1802, in the midst of tensions leading up to Marbury v. Madison, Democratic-Republicans *undid* the Federalist Judiciary Act of 1801. Actual tribunals created by the Federalist Party were eliminated. This forced judges on the Supreme Court to circuit ride again through 1869 (Glickstein 2012; Carpenter 1915; Turner 1965).[25]

In the early 20th century, however, it was the Commerce Court of the United States that was canceled. This body had been created through the 1910 Mann-Elkins Act, staying viable for three years. It was meant to relate specifically to rulings handed down by the Interstate Commerce Commission. Although evidence was not found that the Commerce Court actually favored railroads, this was a live question for many: Dunn (1913) does not find dispositive evidence that railroads were favored by the Court, but this was a question; Lorch (1966) also admits pro-railroad accusations were leveled against the tribunal. George Dix has admitted that, in a separation of powers system, it was jarring to see the destruction of the Commerce Court.

Additionally, this more extreme departmentalism is in evidence, even as recently as a few decades ago. 1982 saw the birth of the United States Court of Appeals for the Federal Circuit, constituted unlike any other federal tribunal based on subjects considered,

---

[23]　*Patchak v. Zinke 2018*, (Roberts, C.J., dissenting), quoted in Aziz Z. Huq, "Judicial Independence and Its Enemies", *Northwestern University Law Review* Vol 115 (2020): Congress had "arrogated the judicial power to itself", p. 2.

[24]　Monaghan emphasizes the broad defense by Justice Thomas of jurisdiction stripping, which in Patchak II reads *United States v Klein* as leading to an interpretation of *Ex parte McCardle* according to which no doubts about the constitutionality of Congressional pushback of this kind remain.

[25]　see https://www.fjc.gov/history/timeline/8276 (accessed on 7 September 2021).

as opposed to geography. Crucially, in the legislative construction of this court, what was *abolished* was the Court of Customs and Patent Appeals as well as the US Court of Claims (Petrowitz 1983). The judicial officers who served on those two bodies were reassigned to the Court of Appeals for the Federal Circuit. A new US Court of Federal Claims was also instituted. In the creation of the Federal Appeals Court, the issue was standardizing rulings at the appellate level. Business leaders supported the changes in light of the importance of uniformity in federal tax disputes Dix (1964) affirms that abolition was not normal politics. Most relevant is that the law involved not just the creation of new courts but the demotion of an Article III tribunal to a non-Article III body. This constitutes further evidence for the reality of jurisdiction limitation, as discussed above (Katz 1930, pp. 903–04).[26]

## 6. Religious Freedom Restoration Act

The most graphic illustration, however, of an existing Congressional ability to assert itself against the courts is found in the ongoing struggle between Congress and the Supreme Court on the question of religious liberty. For decades, even a liberal Court sided maximally with those who claimed that their liberty of worship had been unfairly and unconstitutionally curtailed, culminating in the Sherbert test standard.[27] However, in Employment Division v. Smith, a conservative majority ruled against these specific free exercise protections. The case involved Native Americans who had been denied the right to smoke peyote by state law. Writing for the majority, Justice Scalia held that, when laws are applied in a neutral way, there is no violation. General restrictions on peyote could, therefore, be upheld, even if one of the consequences was a group denied the ability to incorporate it into a sacred ceremony.[28]

In the Religious Freedom Restoration Act legislation of 1993, Congress responded by overruling the Court and essentially making its former *Sherbert* test, upholding the free exercise of religion, law. Congress applied these protections at both the federal and state levels. Subsequently, in 1997, in the case of Boerne v. Flores, the highest US tribunal ruled that Congress *cannot make the Supreme Court's own former standards state and federal law in this way*.[29] The case involved an Archbishop who sought a permit to expand a church in the city of Boerne, and who sued the city upon denial. In its opinion of upholding the permit application rejection, the Court argued that Congress cannot offer its own interpretation of the 14th Amendment. Conceiving of its power in broader terms than *Cooper v. Aaron*, the decision by the Kennedy majority invited the assessment that judicial supremacy was stronger than ever (Lemieux 2017, p. 1074).[30]

Crucially, however, the Court affirmed RFRA at the national, though not the state, level. Congress' own overrule of the nation's preeminent judicial body had been sustained in part. Can a decision of this kind truly represent a reaffirmation of judicial supremacy? As Scott Lemieux has further pointed out, a subsequent case saw the Supremes apply the RFRA standard to strike down Affordable Care Act regulation requiring nuns to provide contraception coverage—a mandate that could have been upheld based on *Employment Division v. Smith*.[31] What about in the years 1993 to 1997? Did judicial supremacy cease to exist?[32] Interestingly enough, as it turns out, a powerful reminder of the ongoing relevance of departmentalism has surfaced even in the context of Supreme Court assertions of supremacy more robust than those found in *Cooper v. Aaron*.

[26]　The power to transfer such jurisdiction to a legislative court had not been expressly recognized until the decision of the *Bakelite* case. On the contrary, the previous decisions and dicta pointed in the opposite direction. See also (Petrowitz 1983, p. 558), fn 135–37, for mention of the Court of Claims converted by Congress to an Article III and back to a non-Article III court.

[27]　*Sherbert v. Verner* 1963.

[28]　*Employment Division v. Smith* 1990.

[29]　*City of Boerne v. Flores* 1997.

[30]　*Cooper v. Aaron* 1958 and *City of Boerne v. Flores* 1997: "If Congress could define its own powers by altering the Fourteenth Amendment's meaning, no longer would the Constitution be 'superior paramount law, unchangeable by ordinary means'".

[31]　(Lemieux 2017). As Lemieux points out, nobody was really affected—see p. 1075.

[32]　This builds on the point of Lemieux 2017: you wouldn't expect to see this challenge in a supremacy regime, pp. 1074–75.

It is easy to see, based on these little-reported facts about the major relevant decisions and the cases of effective legislative override of the courts described above, how Lemieux could think it is departmentalism that straightforwardly exists (Kabala 2020). Consistent with this view, he openly calls for the expansion of the Supreme Court (Lemieux 2018). Lemieux also dismisses a body of literature asserting that, at least since the time of *Cooper v Aaron*, a regime of judicial supremacy has been in place. It represents mere theorizing, for Lemieux, and does not take into account the fundamental fact that departmentalism has always been the reality and continues to exist today in the US.

Yet, at the end of this section that has explored ways in which an effective notwithstanding clause in US politics may exist, it must be acknowledged that the near consensus of legal theorists and experts is that it does not. Instead, the majority view is that judicial supremacy, or the Court effectively having the last word in constitutional disagreements, is the reality. To varying extents, this has been echoed by Larry Kramer (who does not favor judicial supremacy) (Kramer 2012), Barkow (2002), Schauer (2004), Friedman (1998), and many others. Their interpretation is supported by the words of Justice Rehnquist himself (Quoted in Post and Siegel 2004, p. 1039). It just does not appear normal, one might add, for one of the branches of government in a regime of departmentalism to consistently make strong claims of supremacy.[33]

To make his case that the majority view does not matter, Lemieux mentions Keith Whittington, who has moments of skepticism about the normative value of judicial supremacy, and who acknowledges that it has a political foundation. Lemieux brings him up in support of the point that judicial supremacy is a political construction and, presumably, can be undone. However, Whittington explains that this political foundation is very deep, and that it has been a long time in the making. These are points Lemieux does not address (Whittington 2007), which makes his dismissal of this body of literature seem weak. Does it have no valuable insights to offer?[34]

An effective legislative override of the Courts, in a real sense, does exist. It may not be a groundbreaking power. However, how else to explain the instances of stripping, court abolition, and outright rejection of Supreme Court rulings described above? In a real sense, however, a different set of facts also applies. In addition to the abovementioned literature, judicial supremacy is evidenced by the fact the Congress has not successfully applied jurisdiction stripping in the context of a controversial public issue, and that no district or appellate court has been abolished in more than 100 years. This leads to an unnerving question: does the effective power of a legislative override of the courts in the US somehow exist, and not exist, at the same time?

### 7. Two Kinds of Power and Hobbes and Spinoza

A philosophical disagreement about the nature of power sheds light on this dynamic. This is the distinction between potestas and potentia. Various commentators have relied on these categories to describe both Hobbes and Spinoza,[35] and a key feature of this contrast is that the first kind involves legal or juridical influence and authorization, whereas the second depends on direct, concrete, or actually exercised power. This is not to say that other categorizations of influence do not have relevance—take the three faces of power, for example (Lukes 2021). Additionally, the same words have been used in other contexts to denote the opposite. Thus, in commentaries on Aristotle and Aquinas, it is potentia that refers to a potential power, whereas actus involves actualized influence (Freddoso 2015).

---

[33]  Rehnquist; Cooper v. Aaron; Boerne; one way is to actually see dialectic . . . think of Robert Post; Reva Siegel argument (par).

[34]  Kabala, Boleslaw Z. interview with Christopher Wolfe, 14 June 2021. If departmentalism does exist as Lemieux argues, Congress had *overturned* the 1990 Supreme Court decision. If it is able to do this in 1993, why not simply pass a slightly modified equivalent of the 1993 law after the Court's 1997 ruling?

[35]  The literature is vast—for a few recent examples: C Nicco-Kerinvel, "Puissance et individu chez Descartes, Hobbes et Spinoza"; C Altini, "'Potentia' as 'Potestas' An Interpretation of modern politics between Thomas Hobbes and Carl Schmitt"; "Hobbes or Spinoza? Two Epicurean Versions of the Social Contract"; ER Bodde, "Benjamin and Spinoza: Divine Violence and Potentia"; and J Glidden, "Immanence, Potestas, Potentia: How Desire is Produced in Spinoza".

Potentia-potestas has also historically been used by Boethius to describe states of the soul (Dane 1979). It has been applied in contemporary social science to describe outcomes in education (Moss et al. 2018). In contemporary critical and post-modern theory, especially in the writings of Balibar and Deleuze, this contrast additionally refers to the contrast between authorized and actualized power (Deleuze 1992, 1997; Balibar 1998).

Interestingly, these two ways to understand impact translate into the different approaches taken by two 17th century thinkers who are often compared and contrasted. Traditionally, Hobbes—especially in the tradition of post-Marxist Spinoza thinkers—has been presented as an expositor of *juridical* power. Spinoza, on the other hand—and especially in this tradition—has been presented as expounding the other view of concrete power or *potential* (Matheron 1985; Negri 2003). This is evident in both his *Theologico-Political Treatise*, where Spinoza equates the right of nature with the power of nature, and in the later *Political Treatise* (Spinoza 2001, pp. 179, 21, 25).[36] In fact, it is in this latter text, the *Political Treatise*, that the Dutch theorist of democracy adopts a straightforwardly Machiavellian perspective of power *on the ground* (concrete power)—the contract that he had theorized in the TPT now drops out.

Work on potentia and potestas in Hobbes and Spinoza has recently been expanded by Sandra Leonie Field. Interestingly, in building on Tuck and others who see Hobbes as a radical democrat, she makes the case that, by the time of *Leviathan*, Hobbes also conceptualizes popular *potential* (Field 2020). To be more precise: it is not that Field argues Hobbes did not use the word potentia in pre-Leviathan works. Rather, he did, but in the earlier usage denoted above, which differentiated between potentia (potential) and actus (act). Field makes the case that Hobbes came to embrace popular potentia as actualized popular power as he also shed his Scholastic views on natural science.[37] The new, experimental knowledge brought together potentia and actus, but in a way that introduced a possible disjunction between popular potentia and potestas. Popular and effective sovereign power, by the time Hobbes reaches *Leviathan*, is relational and diffused in various networks.

Indeed, the reason that Hobbes, for Field, disfavors democracy in the later books is precisely his appreciation of concrete popular power in its on the ground dynamics. This non-idealized, *actual* influence of the multitude, grasped in its real contours, and if not continually constituted by institutions and constitutional mechanisms in order to sustain a democratic order, produces power blocks. These can lead to oligarchy. Additionally, they can subvert the entire social contract.[38]

Potentia is synonymous with pouvoir constituant. This understanding of popular sovereignty became especially pronounced around the time of the French Revolution (Kalyvas 2005, p. 225), in the thought of the Abbe Sieyes,[39] Tom Paine (Paine 2011; Kates 1989), and Thomas Jefferson (who proposed a constitutional convention every twenty years) (Israel 2019).[40] It is concrete and actualized, the opposite of the top-down command that potestas implies. Pouvoir constituant refers to the power of the people before a constitutional order is in place. On some accounts, as discussed below, it can re-assert itself. This could be in a convention, for example, if the constituted order drifts.

After the preeminent role it played in the French and American revolutions, pouvoir constituant was also reflected in a wave of thinking in the twentieth century. It is typically associated with Carl Schmitt, especially his book *Political Theology*, where the idea is to push back against a sterile legal order associated with the theorist Kelsen. Pouvoir constituant, on this account, is a primordial power of the people that irrupts into everyday affairs. Others writing on the subject in the 20th century include Costantino Mortati (1972), Ernst-

---

[36] Henceforth TPT 177.

[37] Ibid., pp. 49–54.

[38] Ibid., pp. 101–16.

[39] (Sieyes 1789).

[40] See Thomas Jefferson to William Smith, Paris 13 November 1787. https://www.loc.gov/exhibits/jefferson/105.html#:~:text=I%20say%20nothing%20of%20it\T1\textquoterights,%2C%20%26%20always%2C%20well%20informed (accessed on 7 September 2021).

Wolfgang Böckenförde (1991), Friedrich (1950), Heller (1971), and Hannah (1972, 2006). Today, one thinker who has called attention to the concept and deplores what he sees as its insufficient application in politics is Andreas Kalyvas. In the argument to follow, his categorization is especially important (Kalyvas 2005).

A number of these theorists, including Schmitt, saw Spinoza as *obscuring* the sources of constituent power (Schmitt 2008). However, there is even some evidence that, before 1938, Schmitt himself was not committed to this view (Koekkoek 2014). Additionally, recently, the case has been made—especially by Filippo del Luchese—that Spinoza has a special and privileged place in the attempts to understand pouvoir constituant (del Lucchese 2016). On a plausible reading, Spinoza is a theorist of this non-normative power, which emphasizes immanence as opposed to top-down control, just as he has contributed to conceptualizations of potentia in history.

To the extent that potentia and pouvoir constituant occur in the judicial review legal literature, they refer to a power of the people that, if in evidence, could impact constituted institutions. One occurrence is in a French article, broadly critical of the idea of popular constitutionalism (Fassassi 2008). Discussing its different versions in American politics, from the total rejection thereof (Mark Tushnet) to a qualified disagreement with judicial supremacy still affirming review (Larry Kramer), this piece faults popular constitutionalism for not protecting minority rights. It allows that people *themselves* might support a regime of judicial supremacy, if one is to judge by popular unwillingness to overturn Warren and other courts' edicts in the last few decades. Another mention of pouvoir constituant as supporting popular resistance to judicial rule occurs in an article discussing the work of Agamben in relation to radical democracy (Borislalov 2005).

However, these terms can also apply in reverse, further deepening the ontological ambiguity of the power in question. If the "orthodox" literature is true, then it is a notwithstanding clause of sorts in the US presidential system that exists on paper, as potestas, and that has only appeared as glimmers of potentia. It is the power to veto the courts, in other words, that exists primarily on paper. It is a mode of resistance against the judicial elite that the people have not yet applied successfully. The rule of judges, on the other hand, can point to its influence as potentia: concrete and direct, with effective jurisdiction stripping still isolated and not involving major constitutional issues. That this inversion of the categories is possible is recognized in a recent article on parliamentary and judicial supremacy, which speaks of the courts usurping pouvoir constituant and, in effect, becoming the actualized power in the polity that is potential (Danchin 2017, p. 29).

Strikingly, in a broad sample of articles defending both supremacy and departmentalism, this notion—pouvoir constituant or potentia—is hardly mentioned. This is true of Whittington's articles, but also of Kramer's and Christopher Wolfe's, and, it turns out, of those of several other prominent theorists (Hodges 1958), A few examples include Ides (1999), Hodges (1958), Whittington (2001), Mendelson (1947), and Fleming (2005).[41] The argument presented by Carlos Rios suggests that pouvoir constituant, in an international context at least, receives attention. However, clearly, there is a conceptual hole in the literature on American constitutionalism, given the direct relevance of these two conceptualizations of sovereignty, which Hobbes and Spinoza can help us to explore, to the dynamics of an effective notwithstanding clause (departmentalism) and the judicial supremacy that will not tolerate it.[42]

## 8. Hobbes and Spinoza on Institutional Courts

The potestas/potentia contrast also provides additional clarity on the specific ways in which Hobbes and Spinoza set up their courts. Their different institutionalizations of the judicial branch are noticeable and make sense in terms of the competing conceptualizations of power under consideration. Predictably, Hobbes isolates his judges from popular control.

---

41  The full list is too long to reproduce in this footnote—the point is that a number of articles that could well tie the subject back to different views of sovereignty do not.

42  (Colon-Rios 2014a) see footnotes 32, 33 (p. 153) and 68 (p. 163).

He also authorizes judicial officers as subordinate to the formal power (potestas) of the *Leviathan*. Spinoza, on the other hand, keeps judicial officers more closely connected to the people, consistent with a view of popular power that is potentia.

The basic structure of sovereignty as potestas that Hobbes theorizes in his major works remains unchanged from *The Elements* (Hobbes 1984) to *De Cive* (Hobbes 2014) and *Leviathan* (Hobbes 1651). In all of these, a separation of powers is cautioned against in the strongest of terms, and there is not supposed to be a possibility of judicial review or of a judge overruling the Sovereign. Only in *Leviathan,* however, is there explicit language about the authorization of judges by the Sovereign, and, indeed, their representation of him. This may not be surprising, given that it is not until *Leviathan* (as opposed to the two earlier works) that Hobbes introduces his theory of representation. Additionally, as is not the case in *The Elements* and *De Cive*, judicial officers appear in the text as a distinct category of public servants whose qualifications need to be considered separately from those required of other kinds of public officials. Hobbes's judges can presumably serve for life in all three of his major works, and he does not provide guidelines on how long judges should stay in a position. However, in *Leviathan*, where, unlike in *The Elements* and *De Cive*, Hobbes does provide specific qualifications for judges, one of them mediates in favor of a longer, rather than shorter, tenure: this is due to their familiarity with the law, gained through long study in a secure position.[43]

The contrast, Spinoza is instructive. This is especially clear in the book in which potentia is highlighted to a greater extent than potestas, the *Political Treatise.* Spinoza moves from regime to regime, kingship in chapter five through three different kinds of aristocracy in chapters eight to ten, to democracy in the final and unfinished chapter eleven. As he does so, each successive governmental type is democratized.

In the kingship, Spinoza provides a set of instructions that the Hobbesian Leviathan does not:

> These judges, too, should be *very many* [italics mine], and their number should be odd: E.g., sixty-one or fifty-one at least. No more than one judge should be appointed from each clan, and not for life. Here, again some portion of them should retire every year and an equal number of others be appointed. These should be from other clans and should be aged over forty

> (Spinoza 2000).[44]

The Hobbesian qualification of familiarity with the law (that mediates in favor of long tenure) is also not mentioned.

The same holds true in the first kind of aristocracy: "With regard to the number of judges, however, a consideration of this kind of constitution does not demand any special figure; but, as in the case of monarchy, it is of prime importance to see that the judges are *too numerous to be corrupted by a private person* [italics mine]".[45] As in the monarchy chapter, Spinoza also makes clear that the judges are *not* to serve for life, which, to repeat, Hobbes never does.[46] Additionally, judges are selected from the ranks of patricians, *by the supreme council itself* [italics mine].[47] The discussion of an aristocracy with multiple capital cities in the next chapter has a brief mention of judges to the same effect, with the recommendations of the previous chapter by all appearances retained in individual townships.[48] Judges are also not referenced a single time in the next portion of the PT, which contains final thoughts

---

[43]  Hobbes (1651). *Leviathan.* London: Gutenberg edition): https://www.gutenberg.org/files/3207/3207-h/3207-h.htm (accessed on 7 September 2021)/Part II Chptr 26, "The Abilities Required of a Judge". Henceforth L 26.

[44]  Henceforth PT 72.

[45]  Ibid., p. 114.

[46]  Ibid.

[47]  Ibid., p. 115.

[48]  Ibid., p. 125.

on aristocracy. Additionally, they are not mentioned in the final, and unfinished, discussion of democracy that is chapter 11 of the work.[49]

However, the third aristocracy chapter in the PT highlights the difference between Spinoza and Hobbes's judicial recommendations in yet another way. It contains mention of an office Spinoza refers to as the "syndics". What do these do? Spinoza analogizes them to the dictatorship in the Roman republic; what is at stake is a high-powered, if temporary, executive. He describes the "dictator" (syndics) as preserving the *form* of a commonwealth,[50] which may involve preventing movement that leads to corruption (Pocock 1975).[51] However, the point, for Spinoza, is who is *not* ensuring fidelity to this constitutional form, is judges. It is very unlikely that he envisions a robust or strong form of judicial review given this description. Although Hobbes does not either, an out of the way phrase in *Leviathan* is significant: Hobbes distinguishes between "sovereign" and "subordinate" judges.[52] One way to interpret this distinction is that it is possible, for Hobbes, for the Sovereign himself to act as a judge. In Hobbes but not Spinoza, then, the possibility of a judge preserving the form of the commonwealth is apparent.

In terms of the actual set-up of the courts, a possibility of effective empowerment of judicial institutions that leads to a co-optation of sovereignty also becomes apparent in Hobbes, but not in Spinoza. How so? The possibility is related to how a court might become an independent power center, even going so far as to claim judicial supremacy. To recall, Field's argument is that Hobbes's embrace of actualized popular power (potentia) is greater in the later *Leviathan* than in his earlier works. *However, Hobbes sees the contours of actualized popular power in such a way that oligarchy, or power centers that undercut sovereignty, spontaneously emerge.* Again, this is why, in *Leviathan*, Hobbes emphasizes to the greatest extent the repressive machinery of his state. Here is Field:

> . . . Hobbes shows a new and persistent concern with these informal associations: worrying about eminent individuals, masters with too many servants, the immoderate greatness of towns, and the accumulation of treasure by monopolies or farms (L 372–373, 460–461, 516–517, 22.31, 27.15, 29.19, 29.21). He also inserts a new anthropology of religion . . . worrying that new religions can resurge at any time (L 164–171, 176–179, 180–181, 12.1–11, 12.20, 12.23). *Any de facto social groupings are of concern: whether familial, socio-economic, or religious* [italics mine] (L 372–373, 22.32).[53]

In considering these categories, note some of the groupings Field mentions: towns and monopolies. Both of these, to exist in *Leviathan*, need to be authorized by the Sovereign.[54] If *they* are able to gain in effective power in a way that takes away from authorized sovereignty, it is possible that an authorized class of public officials in government—including judges—may as well.

Based on this analysis, it may be that Leviathan—in which Hobbes wishes to avoid judicial review, and in which potentia is privileged to a significantly greater extent than is the case in *De Cive* and in *The Elements*—is the last place where the author would want to formally authorize an institutionally distinct class of judges, who, it appears, may receive life tenure.[55] The broader lesson, generally, may be that those who do not favor the supremacy of the courts should not design judicial institutions based on formal grants of power. Instead, they should seek attunement to the dynamics of actualized power (potentia) that govern these tribunals in their interactions with other actors in the life of the Commonwealth.

---

49 Ibid., pp. 135–37.

50 Ibid., pp. 129–30.

51 Though, of course, as is not the case in the Aristotelian versions of republicanism that Pocock presents, Spinoza envisions the executive maintaining constitutional form—thereby aligning himself with Machiavelli.

52 L 26.

53 (Field 2020, pp. 100–1).

54 L 22.

55 (Field 2020, pp. 101–6).

### 9. Degrees of Departmentalism and the In-Betweenness of the Effective US Notwithstanding Clause

The above benefits of drawing Hobbes and Spinoza into judicial discussions are not the only ones. As is discussed in the current section, one of the persisting problems in political theory is also whether pouvoir constituant continues to exercise influence once a political order has been constituted, or whether, by that point, it is irrelevant.[56] On this technical question as well, a Spinozist metaphysics provides additional insights. This, in turn, allows the Hobbes–Spinoza comparison of different kinds of power to provide further clarity on court-related subjects.

Does immanent power have a normative dimension, which allows it to maintain contact with constituted institutions *after* a founding? George Lawson discusses constituent power in chapter two of his major work,[57] making clear that, even though it is pre-political, the popular sovereign is still bound by basic norms that include justice.[58] This idea receives further support from Thomas Hobbes,[59] who, in at least one important interpretive framework, theorizes the original law-giving community as *actually* constrained by natural laws that focus on justice, covenant, and gratitude.[60] Additionally, Hobbes is interesting here: despite that initial contact of people with ethical standards, it is subsequently lost: the sovereign is *completely* asleep (Tuck 2016).[61]

Kalyvas, in responding to answers to Arendt's questions about the viability of pouvoir constituant, further demonstrates the idea of pouvoir constituant in contact with the morality of constituted institutions,[62] and on this overlaps with Böckenförde.[63] Martin Loughlin is an additional example of continuity (Loughlin 2014; del Lucchese 2016, p. 184), as is Rios (Colon-Rios 2014b).[64] For Kalyvas, it is modern theorists such as Schmitt who hold the opposite notion, which involves creation ex nihilo and no contiguity between originary popular sovereignty and norms.[65] Independent confirmation of the possibility of pouvoir constituant that does not occupy a normatively empty space is found in Agamben's comparison of Schmitt and Benjamin. Interestingly enough, in disagreement with Kalyvas, Agamben sees Schmittian originary power still possessing a normative dimension. He admires Benjamin precisely because his state of the exception in no way touches juridical order (Agamben 2005, pp. 54–55).

However, how can primal sovereignty maintain any juridical connection?; is the Benjaminian truly anomic exceptional space not the only logical view? Del Luchesse believes that it is, in fact, Spinoza who explains possible contiguity, with Costantino Mortati (unwittingly) following.[66] The idea is to allow for normative facts, and Spinoza provides a concrete account showing how the conceptualization of these is possible.[67] Key moves include the ontologization of history,[68] as well as Spinoza's vivid use of historical examples, including the travails of the ancient Hebrews.

However this may be, three options seem to emerge. One posits pouvoir constituant staying active with no relinquishing of control. Another envisions it ceasing to exist,

---

[56] (Kalyvas 2005 and del Lucchese 2016).

[57] Lawson, George. *Politica Sacra & Civilis,* or, A model of civil and ecclesiastical government wherein, besides the positive doctrine concerning state and church in general, are debated the principal controversies of the times concerning the constitution of the state and Church of England, tending to righteousness, truth, and peace (London: 1689; Ann Arbor: Text Creation Partnership, 2011), chp. 2, pp. 17–18, https://quod.lib.umich.edu/e/eebo/A49800.0001.001/1:6?rgn=div1;view=fulltext (accessed on 7 September 2021), see (Kalyvas 2005, p. 226).

[58] L 14 and 15.

[59] Kalyvas (2005) does not mention Hobbes.

[60] L 15.

[61] (Tuck 2016).

[62] (Kalyvas 2005, pp. 230–37).

[63] Ibid., p. 234.

[64] Mentioned by del Luchese p. 183–see also: https://link.springer.com/article/10.1057/s41296-021-00467-z (accessed on 7 September 2021).

[65] (Kalyvas 2005, pp. 227–28)

[66] (del Lucchese 2016, p. 194).

[67] Ibid.

[68] Ibid.

for all practical purposes, so that it never again touches the day to day processes of instituted government and norm-embodying public life (in which case it does not make sense to consider "awakening", in a Jeffersonian constitutional convention, of the popular sovereign).[69] A third views pouvoir constituant staying active *to a degree.* This is suggested by Kalyvas' language itself: he refers to popular sovereignty that will "wane"[70] into the time after the founding moment.

As it turns out, the question of degree is addressed by Spinoza in an interesting way that applies. Seeing how requires a brief trip through the *Ethics*, where Spinoza distinguishes between natura naturans (de Spinoza 1994)[71] and natura naturata.[72] A rough English translation is "nature naturing" and "nature natured", respectively. In terms of the framework in Book I of the *Ethics*, nature understood in *both* the first and second ways is God, the one existing substance. However, this substance has infinitely many attributes,[73] with thought and extension the two that human beings can access.[74] There is also a level of reality that is more particular: individual cogitations are modes of the attribute of thought, and individual objects occupying space, say, a tree or a stone, are modes of the attribute of extension.[75]

Crucially, natura naturans, or the attributes of thought and extension, are characterized by the production of effects through their nature alone.[76] Natura naturata, or individual thoughts and objects, on the other hand, are conceptualized profitably as part of a chain of causality without reference to their nature alone.[77] These individual modes follow from multiple factors, including external ones that impact the individual mental or physical event. It is easy to see, in this sense, how natura naturans, or the attributes of thought and extension that include the laws governing them, produce individual ideas and bodies. Natura naturans is active, constituting and generating natura naturata, which consists of these individual objects, and which are, therefore, passive.[78]

The power of any of these modes is the ability to persist, Spinoza's famous conatus doctrine.[79] Power, for Spinoza, is synonymous with activity, insofar as power *is* the power to act, and to repeat involves producing effects only through the nature of whatever is considered powerful and active, which is another way to indicate what Spinoza sees as adequate causation.[80] The active mind has what Spinoza refers to as adequate ideas; an active body is characterized by positive emotions, which lead not only in the direction of clear causality but to the greater ability of the body to persist.

Additionally, an active individual, understood as either a powerful body or mind, can also be said to be in possession of his or her right. This category ends up being very important for Spinoza. Do we understand that individual, in other words, as the mental or physical effects associated with and traceable back to said person as cause, or to said person in combination with other causes? The active individual is generative as a cause, analogously to natura naturans but not natura naturata.[81]

This may sound abstract, but connections between these metaphysical categories and institutions have already been made in the social sciences in a way that points to the courts.

---

69  (Tuck 2016).
70  (Kalyvas 2005, p. 234).
71  EI P29 S1, pp. 104–15. Henceforth EI P29 S1.
72  Ibid.
73  EI P11.
74  EII P1 & P2.
75  EI P16, P23 & P25Cor.
76  *Ethics.*
77  *Ethics.*
78  *Ethics.*
79  E III P6 & P7.
80  E III D1.
81  Although Spinoza's individual is a mode, and there are only two attributes—thought and extension—accessible to human beings, there are indications in his work that different kinds of modes exist, and that some may be more like attributes than others. The question of whether the difference between modes and attributes itself is a matter of degree is not one to avoid indefinitely. See (Melamed 2015, p. 22).

Thus, Douglas J. Den Uyl, Spinoza scholar of note, emphasizes one of James Buchanan's appendices in *The Calculus of Consent*. Here, the architect of public choice economics was willing to see ideals as natura naturans, and social realities constituted by them as natura naturata (Den Uyl 2014). There are multiple possible configurations in which these categories are interchanged, especially given that they are not fixed or essential.[82] Thus, what is natura naturans in one frame can become natura naturata in another. One example Den Uyl gives is of Buchanan's "RAA's", or "Relatively Absolute Absolutes". Different levels of rules in the game of golf can be understood differently, depending on one's context. The rules themselves can stand as "natura naturans" in relation to specific tactics that golfers apply; or, if the United States Golf Association convenes to deliberate and change those rules, its meeting has taken on the function of natura naturans, whereas the resulting rules are now modes or natura naturata.[83]

Where the connection to pouvoir constituant and a people capable *to a degree* of pushing back against the courts becomes especially clear, is in the robust connection of power in the *Ethics* to an important concept for Spinoza in the *Political Treatise*,[84] widely discussed in the literature (Steinberg 2008; Santos-Campos 2014; Piirimae 2002), which is being in a state "of one's own right" or maintaining control of one's right (sui juris). This applies not just to individuals but to social bodies.[85] Simply put, for a social body to have the characteristic of activity as opposed to passivity, or to have power, is also to have possession of its rights.[86] For a social body, this can involve either producing effects or—as is true of natura naturans—generating them in a way clearly caused by its nature alone.[87]

Without prejudging any particular set of circumstances, then, it is certainly conceivable in a regime of judicial supremacy that a multitude that has to do as the Court bids and is not able to engage in dialogue with the Court *is not in possession of its own rights.* In other words, this would suggest that the multitude knows that, as a result of the Court, its power to interact with the policy environment (acting and being acted upon) is limited. It is powerless to generate causes based on its nature alone, and it knows *there is no context in which it can be seen as natura naturans.* It is beholden to an external body (the Court). Additionally, this is especially true if the judges are not intervening for the sake of upholding the rule of law or have become an unresponsive elite. An *active* people, one in possession of its right, in this context, is able to abolish courts and reconstitute the judiciary at will.

However, and it is Sandra Leonie Field's accomplishment to point this out, to demonstrate power and stay in possession of one's right (sui juris) is not a simple binary for Spinoza; it is a matter of degree. Sandra Field provides several examples, including acting according to one's nature so long as one has food and drink, or a crisis event does not intervene from outside. In making the general point, she goes against important arguments in the literature, which have seen activity and passivity as mutually exclusive (Steinberg 2008).[88] What this means is that if pouvoir constituant is understood as the power of the people that can impact the courts, it, too, can exist as a matter of degree.

Therefore, it is not a question in the US of whether an effective notwithstanding power does, or does not, exist. Rather, it is a question of the degree to which it does (allowing the people to be independent from or in possession of right relative to the Court). This helps us to make sense of Kalyvas' suggestion that pouvoir constituant waxes and waves ever after a founding.

Additionally, this, of course, makes further sense of the departmentalism literature. Post and Siegel point to different departmentalist views, ranging from the position that

---

[82]    Ibid.

[83]    Ibid., pp. 177–78.

[84]    PT 41–42 and 61 (see Field 2020, p. 193).

[85]    (Field 2020, p. 193).

[86]    PT 61.

[87]    Ibid.

[88]    see Field (2020, p. 194).

judicial review is not at all acceptable, to a more limited version in which the Court can make a decision about one set of circumstances without drawing out a general principle that will also apply, in future litigation, to similar situations (Post and Siegel 2004). Mike Paulsen, of course, argues that any Court decision applies only to the particulars before the justices, which is not a position that all departmentalists or those in favor of more effective vetoes of the judicial branch need accept (Paulsen 2000). Additionally, the importance of degree in conceptualizing popular sovereignty that resists a judicial elite makes sense, especially, in assessing our earlier discussion of jurisdiction stripping based on Fallon's important article; depending on the number of different kinds of courts that one believes Congress can deprive of the ability to hear cases, one can say that the effective notwithstanding clause exists to a greater, or lesser, extent. All these considerations are more easily accounted for if the existence of an effective notwithstanding clause in the US is not a binary category but is, in fact, a matter of degree.

## 10. Dynamic Interplay between *Potentia* and *Potestas*

Critically, in support of the theory that we are presenting (that both potentia and potestas need to be taken into account to understand the court situation in the US), Hobbes's as well as Spinoza's texts affirm that *potestas and potentia can operate at the same time*. Early on, Hobbes demonstrates only authorized power. To emphasize, though, in his later works (*Leviathan*), he shows evidence of concrete popular sovereign control, combined with the social contract (and therefore co-existing with the formal grant of power made by those in the state of nature). The point is that, by the time of *Leviathan*, the English political philosopher sees *both* kinds of power as in need of simultaneous consideration. Additionally, Spinoza, early on, in the *Theologico-Political Treatise,* focuses more on a social contract (formal or authorized power).[89] This contract *drops out* of the *Political Treatise.*[90] However, in that later book, as we show presently, the direct or Machiavellian application of power is *also* combined with formal grants of power (authorized influence).

Where are the formal grants of power evident in the *Political Treatise?* This has to do with Spinoza's constitutionalism, which is lacking in Hobbes (Prokhovnik 2001; Skeaff 2018; Dyzenhaus 2009).[91] Critically, the question of a social contract is not the same as that of a constitution.[92] Constitutional prescriptions, formal prescriptions for the way power should be organized and diffused in the *Political Treatise,* very much exist alongside the many instances of concrete power—in the chapters on aristocracy, where the institutional checks (and balances) are described.[93]

To be clear, Spinoza is *not* saying, in chapters 6–11 of the *Political Treatise*, that a written blueprint of government has to exist.[94] It can, but even if it does not, Spinoza wants his own recommendations—about the size and number of different bodies—to be authoritative.[95] He wants these written recommendations with regard to offices, rotation, and election to control and to have power. Whether or not the states receiving the recommendations adopt their own written constitution after having implemented Spinoza's textual recommendations, the point is that his formal prescriptions have the status of potestas, not potentia.

This raises the question: why would Spinoza pursue constitutionalism, or the codification of rules (on paper, even if only in his texts), given the overwhelming emphasis on potentia, or direct and *non-codified* power, in his later texts especially? An interesting

---

[89] TPT 173–84.

[90] PT 57–60.

[91] On Spinoza and constitutionalism, see Justin Steinberg, "Spinoza's Political Philosophy" (2008). Although small steps towards a "constitutionalist Hobbes" have been taken in the literature, there are simply fewer of these because Hobbes's intent seems very hard to square with constitutionalism of any kind.

[92] A number of ancient Greek cities, for example, would have had the latter but not the former.

[93] PT 95–134.

[94] PT 64–137.

[95] How else to explain the proliferation of institutional detail in these chapters?

literature speaks to this question, which almost certainly has relevance for those on the left and right who are pursuing a democratic reform of the Court (Newton 2006). The main idea is that potestas may *increase potential* (Priban 2018). To phrase this another way: although there is a possible disconnect between power that is authorized and power that is exercised (Spinoza's critique of Hobbes), nevertheless it is also potentially the case that reification or articulation (written or not) of an ideal aids or increases the direct expression of power.[96]

Another way to phrase it is that the words according to which a power already exists may help it come into being further; a more explicit constitutional provision to the effect that a political step is legitimate may be the needed motivation to push a waffling actor from indecision into action. If this is true, as the literature connecting potestas and potentia in Spinoza suggests that it is, a declaration of Congress and the President, or even a constitutional amendment more explicit than the language of Article III, is worth considering for defenders of a departmentalist notwithstanding power in the United States.

## 11. Conclusions

In this article, we have considered a paradox with respect to the question of whether an American notwithstanding clause involving the power of Congress to overrule the Courts already exists or not. From one perspective, on paper, this power does exist. From another perspective, however, related to the direct application of immediate power, it does not. There has not been the kind of jurisdiction stripping that would provoke a serious confrontation between the Court and Congress. Additionally, it is striking, in this context, that the consensus of legal experts is that, in the US, judicial supremacy prevails. What we have shown is that this paradox can be illustrated through a comparison of two different kinds of power, which are also helpfully illustrated by the kinds of sovereign influence two 17th century philosophers are often read as supporting: potestas when it comes to Hobbes, and potentia as it relates to Spinoza.

First, we described the notwithstanding clauses in both Canada and Israel. Both these countries' adaptation of departmentalism is of recent vintage, not more than 30–40 years old. We described the constitutional provisions on which both parliamentary systems of government rely to make their overrides a reality. We further considered important examples of the application of these overrides in both countries.

Second, we noted that, despite the US Constitution not having a notwithstanding clause, a number of concrete proofs exist that scattered American departmentalism is real. These amount to an effective notwithstanding clause, although in a presidential system. The demonstrated facts include jurisdiction stripping—multifaceted and subtle as well as more significant—abolition of courts, and legislative overrule of Supreme Court decisions. These powers are affirmed by a body of literature that refers to the constitutionality of Congress pushing back against the courts in this way as the "orthodox view". At the same time, and frustratingly, researchers such as Fallon point out that the power to strip jurisdiction is highly context dependent. Additionally, the near consensus of the legal profession states that it is judicial supremacy, not departmentalism, that is the norm. Even those who characterize Congressional pushback against the courts as the "orthodox view", as it turns out, admit that this kind of overrule has not been exercised on significant issues, in a way that would provoke an actual settling of the issues with the courts.

Third, in unpacking the possibility that two perspectives are at work (so that it seems this power to overturn the courts exists, and does not exist), we explore two kinds of sovereign control. These are potestas (authorized) and potentia (concrete). Potentia corresponds to departmentalism as *not* practiced in the US, and potestas refers to the authorization of people to strip courts of jurisdiction. Of course, potestas can also refer to the formal grant of judicial authority, which, however, the Court has provided to itself. These two kinds of power also happen to be described by Hobbes and Spinoza. We

---

[96] Ibid.

presented evidence from Hobbes and Spinoza that potestas and potentia can apply at the same time. Further connections between Spinoza and potentia and pouvoir constituant are in evidence.

Fourth, we showed how, in keeping with these distinctions, Spinoza connects the courts back to the people more directly than does Hobbes. Hobbes relies on a *nomination* strategy, with judicial officers the representatives of the sovereign. Spinoza relies on a more democratic route, essentially selection by lot, insofar as those on the judicial council are selected randomly from tribes with variations across both the kingship and different kinds of aristocracies. The selections are not lifetime, and given the size of the republics he is considering and the numbers for the judicial councils that he proposes, it is clear that Spinoza is comfortably on the side of not allowing judges to become a class apart from the popular sovereign, a position that is certainly associated with departmentalism today.

Fifth, we demonstrated two further clarifications that this potestas–potentia distinction, relying on Spinoza's metaphysics, can provide. For a long time, the debate over potentia or pouvoir constituant has involved the question of whether the latter can irrupt into the political order under normal conditions. However, for Spinoza, another way of saying potentia, or pouvoir constituant, is activity "in possession of one's right". This refers not just to individuals but to institutions and collectivities as well. Additionally, from a departmentalist perspective, of course, it is the multitude able to engage in critical dialogue with the courts that is in possession of its right. Crucially, in applying the criterion of "in possession of one's right", it turns out that the reality of legislative pushback against the courts is a matter of degree. We showed that the constitutional calculus may be different with respect to different kinds of courts, damages v. remedies, injunctive relief, etc. This is also explicable precisely if the power of the people to apply departmentalism is not a question of existence vs. non-existence–but, instead, if its reality is context dependent and a question of degree.

Finally, we showed how, especially from the perspective of what to do, potentia and potestas are not mutually exclusive categories. There is a dialectical relationship, and sometimes *adding* potestas *strengthens* potentia. This is mentioned in the literature. On the level of intuition, it makes sense because power may not be exercised without a formal reminder of existence; looking on the formal authorization may give people additional confidence to *act.* This is especially evident in Spinoza's political treatise, where, despite having done away with the contract, Spinoza codifies the details of many institutional arrangements in his own recommendations for different kinds of regimes. There is not the insistence that a written constitution be adopted. However, what is clear is that Spinoza's own words and prescriptions, which amount to constitutionalism (*potestas*), contribute to *potentia.* For those willing to go the departmentalist route, additional formalization of power may, therefore, result in more power exercised on the ground.

However, what about insight that this potestas¬¬–potentia framework provides to people in court reform debates who do *not* wish to move further in a notwithstanding clause or departmentalist direction? Are we only finding cover for people who wish to resist strong judicial review? This is far from the case. As outlined presently, potestas and potentia, along with natura naturans and natura naturata, can helpfully illuminate reasons to stay with a judicial elite as well.

Can a notwithstanding clause or departmentalism generally lead effectively, in terms *of potentia,* to a majoritarianism that sweeps away the rule of law? For Schmitt, writing in an early book, the raw popular power or pouvoir constituant can extend to the people deposing a presiding judge (Schmitt 2008). Additionally, in the work of Giorgio Agamben, outright abolition of the courts does seem an envisioned possibility. His supreme distrust of the courts valorizes potentia but appears to lead to a call for popular and salvific violence. We explore this at greater length in a future article.

For now, in surveying the history of the state of the exception, it is clear that Agamben does make the case that modern thought generally allows for a suspension of normal rule to prevail. To be clear, this is not a suspension of law by the people who, as subject, can

apply popular power directly. It is the modern state that introduces exceptional conditions, reducing human beings to mere life.

Courts that purport to uphold the rule of law participate in this dynamic, as one article considering Agamben's critique of human rights makes clear. Here is a quote speaking to this dynamic:

> This case [involving undocumented individuals who occupied a church in Paris in 1996] manifests the strengths and limits of Agamben's biopolitical analysis. The Court decision clearly shows how sovereign exception continues to pervade human rights norms, challenging controversial accounts that see rights as normative on sovereign power; freedom of assembly is codified in Article 11 of ECHR; yet, the exception clause of the same article allows states to impose restrictions on very ambiguous grounds, including 'national security' or 'public safety,' which can easily pave the way for arbitrariness (ECHR, Art. 11). (Ayten 2012)

Courts are implicated, in other words. Additionally, the analysis of how tribunals of various kinds through a logic of *rights* can contribute to bio-power is evident in Agamben's view of habeas corpus. In linking courts to the entrenchment of biopower, Agamben calls for a politics beyond human rights and judicial channeling of sovereignty.

This is definitely a politics of immanence, with respect to which connections between his work and Spinoza's have been pointed out (Agamben 1999; Vardoulakis 2010; Bernstein 2017; Vardoulakis 2010; Klein 2003). Interestingly enough, Agamben sees even pouvoir constituant as implicated in oppression, through the inevitable connection to a sovereign state, and so he wishes to go beyond pouvoir constituant in waiting for divine violence that would leave behind any connection to sovereignty. This seems clearly indicated by his sympathy with Benjamin, in chapter four of his *State of Exception* (Agamben 2005, pp. 52–64).

However, does this advance the cause of an active multitude from a Spinozist point of view? Throughout his works, the Jewish–Dutch philosopher considers also the "intellectual love of God"[97], which seems the preserve of a privileged few, and which has invited charges of elitism against Spinoza's works.[98] A multitude that is not beholden to an overbearing judiciary may, in fact, be more "in possession of its right" and thus, to a greater degree, *active* as opposed to *passive*. However, what about a multitude that jurisdiction-stripped or abolished courts in anger? Spinoza does write that "the mob is terrifying if unafraid".[99] If permitted to act in this way, on Spinoza's terms and generally, it seems that a multitude would lose power in the end given the connection of anger to negative and passive affects.[100] If so, however, a notwithstanding clause equivalent or departmentalist proposal to increase the potentia of the people, even if through additional formal authorization, that resulted in the people acting unwisely appears in a different light. One may ask whether it does not actually weaken the people and make them more passive in the long run.

**Author Contributions:** Conceptualization, B.Z.K. and R.J.; Writing—original draft preparation, B.Z.K. and R.J.; Writing—review and editing, B.Z.K. and R.J. All authors have read and agreed to the published version of the manuscript.

**Funding:** This research received no external funding.

**Institutional Review Board Statement:** Not applicable.

**Informed Consent Statement:** Not applicable.

**Data Availability Statement:** Not applicable.

**Conflicts of Interest:** The authors declare no conflict of interest.

---

97    TPT 48–58, EVP24–37

98    EIIIP44, EIIIP59, and EIVP18.

99    EIVP54S

100    EIIIP44, EIIIP59, and EIVP18.

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
