# Peer review of "An American Notwithstanding Clause? Between Potestas and Potentia"

_laws_

Round 1
Reviewer 1 Report
A decent, but rather bloated piece -- less would be much more; the piece cannot seem to decide whether it is a philosophical reflection or a political intervention. It tends to veer to the former, not the latter. Recent developments in Canada are illuminating and worth referencing.
Author Response
Dear Reviewer:
Many thanks for your helpful comments. We have added references (pg. 5) to Doug Ford's use of the notwithstanding clause in Ontario and Francois Legault's reliance on it in Quebec, per your suggestion that "recent developments in Canada are illuminating and worth referencing."
Since the piece does aim to be a philosophical reflection that allows multiple parties to intervene politically in a more thoughtful and civil way, we are not at present removing entire sections of the article for the sake of streamlining, which we feel could lessen its intended interdisciplinary impact.
Expressing our appreciation once more - hope you find this acceptable!
Author
Reviewer 2 Report
Summary:
This article considers the case for introducing a formal override or notwithstanding clause within US public law, in order to increase popular responsiveness and counter judicial overreach.
The author argues that paradoxically, the override power both does and does not exist, a paradox which the author explains by appeal to the early modern distinction between potestas (authorized power) and potentia (concretized power). The override power does exist as potestas but not as potentia. In order for it to exist as potentia, the author recommends additional codification and formalization of the already-existing override power.
Merits:
This article brings a novel philosophical frame to bear on public law debates. In so doing, it parses out the complex senses in which the US does, does not, and might in future have an override clause. Both the distinction between authorized and concretized power, and the notion of power as a matter of degree (rather than an object which either is or is not possessed), do real work here.
For improvement:
The section introducing the philosophical distinctions (p12ff) brings on a lot of different authors and ideas all at once, and sometimes the moves are too quick. (Eg OK pouvoir constituant is assimilated with potentia, but then how does that fit with the claim that for Hobbes, potentia is oligarchic?) Even with the more nuts-and-bolts discussion of Hobbes/Spinoza on the structure of the judiciary (p15ff), I remain a little unclear on what distinctions exactly are in play and how they function in the author's texts, and how strongly distinct Hobbes/Spinoza are in their own thinking of the proper selection of the judiciary. That is, the paper is less strong as textual/philosophical exegesis of the historical texts. But perhaps this doesn't matter as its primary focus as the distinction and lessons it wants to draw with respect to contemporary US public law is clear enough.
Author Response
Dear Reviewer,
Many thanks for your helpful comments.
We are encouraged that whereas one of the readers found the piece to be philosophic rather than political, you interpret it as more directly relevant to public law controversies. Our aim was to produce an article that could serve both as a reflection on deeper underlying concepts, and that could allow for more considered and deliberate intervention in contemporary debates!
With respect to the move that you suggest is critically important, related to Hobbes and oligarchy (p 12ff) - we have added clarifying language on pg. 13. Raw popular power, according to Field's important interpretation of Hobbes & Spinoza, can become oligarchic if an ordered multitude is not continually constituted and sustained by democratic institutions and constitutional mechanisms (of the kind that Spinoza explicitly prescribes - and that we discuss - in "Hobbes and Spinoza on Institutional Courts" (pp. 15-17), and "Dynamic Interplay Between Potentia and Potestas" (pp. 21-22)).
Again expressing our appreciation - hope you find this acceptable!
Best wishes,
Author